# Resveratrol and SIRT1: Antiaging Cornerstones for Oocytes?

**DOI:** 10.3390/nu14235101

**Published:** 2022-12-01

**Authors:** Arkadiusz Grzeczka, Paweł Kordowitzki

**Affiliations:** Department of Preclinical and Basic Sciences, Faculty of Biological and Veterinary Sciences, Nicolaus Copernicus University, Gagarina Street 1, 87-100 Torun, Poland

**Keywords:** sirtuins, SIRT1, oocyte maturation, resveratrol, metaphase II, aging, oxidative stress

## Abstract

It is well-known that there is an enormous variability in the aging-related decline of oocytes’ quantity and their developmental competence among mammalian species. The implication of female germline aging is profound from the perspective of evolutionary conservation of the aging mechanism, a topic of continuous and widespread interest that has yet to be fully addressed for the mammalian oocyte. There is a certain need to develop novel antiaging strategies to delay or slow down aging, or even to reverse the aging phenotype in the oocyte. In the past two decades, several antioxidants have been tested for this purpose. Resveratrol is one of these latter-mentioned compounds, which has shown anti-inflammatory and antiaging properties in a dose-dependent manner. Interestingly, resveratrol appears to enhance the activity of so-called Sirtuin 1, too. Therefore, the aim of this review is to summarize and discuss the latest findings related to resveratrol, Sirtuin 1, and their crosstalk and influence on the mammalian oocyte to elucidate the question of whether these factors can delay or slow down reproductive aging.

## 1. Introduction

There is no doubt that the mammalian oocyte will exhibit lower quality and developmental competence with advancing maternal age. This, in consequence, contributes to increased chances of miscarriage, due to several causes such as aneuploidy, oxidative stress, and loss of synchrony with the uterus at the time of implantation. However, there is an enormous variability in aging-related decline of oocytes’ quantity and their developmental competence among women [1]. The implication of female germline aging is also profound from the perspective of evolutionary conservation of the aging mechanism, a topic of continuous and widespread interest that has yet to be fully addressed for the mammalian oocyte. Thus, it is of high interest to develop methods to delay or slow down aging, or even to reverse the aging phenotype in the oocyte. Numerous antioxidant treatments which have been tested as an antiaging strategy in recent years aimed to improve oocyte and embryo quality, too [2]. One of them was resveratrol, also known as 3,5,4′-trihydroxystilbene (Figure 1). Resveratrol is a phytoalexin and plant polyphenol detected, among others, in red grapes, red wine, and some nuts [3]. Interestingly, it has been reported that resveratrol supplemented at specific low concentrations shows antiaging properties and is able to enhance the activity of so-called sirtuins [3,4]. Furthermore, resveratrol is able to enhance mitochondrial metabolism, due, among other factors, to its impact on oxidative phosphorylation (OXPHOS) and mitochondrial biogenesis. However, it is important to mention that the positive effects of resveratrol are present when this compound is used in low doses [5], since high doses of resveratrol could provoke the inhibition of P450 cytochromes and could induce a mitochondrial-dependent cell death [4,6]. The supplementation with resveratrol as an antioxidant treatment to elevate oocyte quality and in vitro fertilization outcome has been widely investigated in the past decade. Nevertheless, there is no clear consensus with regard to the mechanism of resveratrol’s action on human oocytes, despite their well-known influence on oocyte mitochondria [4]. A detailed description on the interaction between resveratrol and SIRT1 will be discussed in Section 2.

The sirtuin family consists of seven proteins (encoded by the genes SIRT1, SIRT2, SIRT3, SIRT4, SIRT5, SIRT6, and SIRT7), which deacetylate histones in a nicotinamide adenine dinucleotide (NAD^+^)-dependent manner and which are involved in the repair of damaged DNA, too [7]. In addition, the expression of sirtuins can be increased by supplementing a stilbene derivative, i.e., resveratrol, melatonin, astaxanthin, celastrol, or mogroside [8,9,10,11]. The activators presented have been touted for improving oocyte quality or reversing the effects of oxidative stress, and some, such as melatonin, have been proposed as candidates for the composition of media in in vitro cultures. The use of sirtuin activators underscores the importance of these enzymes. SIRT1 is the most thoroughly understood protein of the sirtuin family.

The development, quality, and possibility of fertilization of the oocyte are made possible, among others, by the availability of energy, provided by the mitochondria, and the activity of the cytoskeleton of the egg cell. ATP, which is an expression of the oxidative activity of the mitochondrion, determines the active work of the division spindle—an element of the cellular rack. The functioning of these two elements is tightly regulated by several external, as well as internal, factors so that they can work together. Recently, the sirtuin family has been presented as one of the most important factors influencing precisely these two processes—mitochondrial function and the course of meiotic division [12,13,14]. In particular, SIRT1 is responsible for modulating mitochondrial activity and is one of the protectors against oxidative stress at the ovarian site [15]. In turn, another important aspect of SIRT1 activity is its effect on oocyte age and progressive aging [16]. The antioxidant capabilities of SIRT1 and the pathways it activates help protect telomeres from the accumulating effects of ROS [17]. In addition, recent evidence points to additional pathways for telomere protection and repair in the context of the SIRT1 [18]. This review summarizes the latest findings related to resveratrol, Sirtuin 1, and their cross-talk and influence on the mammalian oocyte to elucidate the question of whether these factors can delay or slow down reproductive aging.

## 2. Interaction between SIRT1 and Resveratrol

SIRT1 consists of N- and C-domain extensions and a catalytic core [19]. The catalytic domain is formed by the Rossmann fold (components: β1, β2, β6–9, and α1, α7, α8, α12) and the subdomain responsible for zinc binding (α10, α11), the former of which is the dominant subdomain in the catalytic core of Sirtuin 1. The latter, on the other hand, which is smaller, is the result of the fusion of two insertions of the origin of the NAD+ linking domain [20]. The substrate that attaches to the active site, between these subdomains, forces a change in the position of the Rossmann fold and the second domain, resulting in the ability to attach to the sirtuin NAD+. This is defined as a transition of the SIRT1 form from open to closed, and the hydrophobic interdomain groove is then observed to disappear, and the minor subdomain binds to zinc [Zn^2+^] [21,22]. Various molecules that modulate the activity of the enzyme can bind to the active site. Inauhzin, a SIRT1 inhibitor that was used to limit tumor growth, bound to the active center via hydrophobic bonds and was positioned deep in the interdomain groove between the active site and peripherally attached NAD+ [23]. In the case of EX527, a potent indole inhibitor of SIRT1, it was indicated that interaction with the active site also contributes to changing the conformation of NAD+, sealing the catalytic pocket, and inducing blockade of acetylated lysine access to the substrate in deacetylation reactions [24]. In contrast, resveratrol allosterically modifies the activity of the SIRT1. Resveratrol has shown to bind to the catalytical core of Sirtuin 1 (Figure 1). In the form of three molecules (Res.1, Res.2, and Res.3), it binds in two domains: Res.1 and Res.2 bind to the N-terminal domain (NTD), and Res.3 binds to the C-terminal domain [25]. It is worth noting that at the NTD site, resveratrol molecules also bind to peptides with which SIRT1 cooperates, where they facilitate, for example, bridging between proteins if only in the case of interaction with P53-AMC [26]. Resveratrol activity also facilitates phosphorylation at the C-domain by promoting the binding of SIRT1 to LKB1 (liver kinase B1), one of the components of biogenesis and mitochondrial respiration [27]. Therefore, resveratrol levels may be translated, via SIRT1, into the development of inflammation, regulation of apoptosis, mitochondrial function, and protection from oxidative stress. Among other reasons, sirtuin has become a therapeutic target in reproductive system diseases [28]. Mitochondrial activity, as well as the efficiency of the organization of cell organelles, is one of the keys to achieving good-quality oocytes. Hence, the use of low-dose resveratrol in disease entities involving the ovaries and disorders related to oocyte development appears to be a reasonable approach. 

## 3. SIRT1 and Quality and Competence of Oocytes

SIRT1 proteins are present in granulosa cells, follicular fluid, and also the oocyte [29,30]. Recently, the detailed temporal and spatial localization of SIRT1 during the maturation of the bovine oocyte was presented. At the germinal vesicle (GV) stage, SIRT1 is mainly localized within the nucleus [31]. Shortly after meiotic division resumes at the GV breakdown (GVBD) stage and as meiotic division progresses, SIRT1 expression levels increase, contributing to the presence of SIRT1 throughout the ooplasm [31]. Although changes in histone acetylation are constantly present during division, their intensity varies between phases, as shown in a mouse model [32]. Of particular interest seems to be the effect of SIRT1 on the course of metaphase I and II [31]. Several works have documented the focus of SIRT1 on a-tubules in metaphase I and II in mice and domestic cattle [31,33]. Accordingly, SIRT1 is involved in meiotic spindle patterning by modifying the level of the histone acetylation (Figure 2) [33]. This fact makes Sirtuin 1 an important activator of metaphase plate formation. Therefore, sirtuin inhibition in a healthy oocyte leads to the formation of abnormal spindles and misaligned chromosomes [34]. In addition, ample evidence supports the disruption of metaphase plaque formation due to oxidative stress [7,35,36]. However, the SIRT1 signaling pathway can restore proper chromosome alignment in states of induced oxidative stress [37,38]. Reversal of division disorders occurs through the pathway between Sirtuin 1 and superoxide dismutase 2 (SIRT1/Sod2), and oocyte quality is improved and allows fertilization. Moreover, SIRT1 inhibition abrogates the positive effects of antioxidants that stimulate the SIRT1/Sod2 axis [37,38]. The SIRT1/Sod2 axis is preceded by the interaction of SIRT1 with the forkhead box transcription factor O3A (FOXO3A), reducing its acetylation [39]. In addition, FOXO3A has a regulatory role in the activation of primary ovarian follicles in collaboration with phosphoinositide 3-kinase (PI3K) [40]. 

By far the most important role in the regularity of oocyte development is played by the surrounding cells. SIRT1 affects granulosa cells mainly through its effect on mitochondria. These organelles are energetic centers and the main reactive oxygen species’ main producers. Depending on the amount of ATP produced, mitochondria can influence most processes within the GC cell and oocytes [41,42]. Silencing of the SIRT1 gene resulted in decreased expression of some of the genes associated with mitochondrial functions (Mitofusin-2 (mfn2); autophagy related 5 (Atg5) and beclin 1), suggesting a regulatory link between SIRT1 and the mitochondrion [43] and contributing to a decrease in cumulus-cell expansion [34]. 

Damage within the mitochondrion can result from diseases in the female reproductive system. Then, within granular cells, mitochondria may undergo swelling, and the damaged cell membrane loses the membrane potential necessary for mitochondrial ATP synthesis [44]. Melatonin supplementation induced an increase in SIRT1-leveled mitochondrial membrane damage through PDK1/Akt activation [44]. In addition, increased body weight, improved hormone levels, and oocyte quality due to resveratrol supplementation have been indicated in rats [45]. 

The level of interaction of progesterone with SIRT1 seems very interesting. Previous reports of a resveratrol-induced positive effect of SIRT1 on progesterone and a negative effect on estrogen bring the thought of SIRT1 interacting with hormones, particularly of granulose origin, closer [46]. This is supported by the close relationship, in postovulatory granulosa cells (in humans the granulosa lutein cells), between SIRT1 expression and activation of the receptor for luteinizing hormone/chorionic gonadotropin (LHCGR) by human gonadotropin (HCG) [47]. In the context of tropic hormones, it is indicated that SIRT1, being a component of the mTOR/SIRT1 axis, may mediate the action of folliculotropic hormone (FSH) [48]. This is evidenced by the increased expression of SIRT1 when added at a dose of 10 ng/mL of folliculotropic hormone. However, this is contradicted by a report of a decrease in FSH levels that occurred with an increase in SIRT1 gene expression [49]. It is also worth mentioning that the expression level of SIRT1 can also be affected by androgen deficiency, causing abnormalities in oocyte development [50].

## 4. SIRT1 and Aging

The multidirectional activity of SIRT1 contributes to changes in the epigenome and mainly in the DNA acetylome, as well as the oxidative status of the cell, and is one of the keys to understanding oocyte aging processes [16,51,52]. A reflection of aging processes is the final sequences of chromosomes, called telomeres. They are mainly responsible for chromosomes’ continued integrity and stability [53,54]. However, with each division, telomeres become shorter due to the shortening of descendant chromosomes within the telomeres and telomerase deficiency, until eventually the coding part of chromosomes is damaged [55]. This process is strictly defined; however, certain factors can disrupt the telomere clock. It has been shown that the accumulation of ROS leads to an acceleration of telomere shortening, thereby accelerating aging. In older oocytes, in addition to increased concentrations of reactive oxygen species, mitochondrial damage, increased meiotic errors, autophagy, and premature apoptosis are noted [11,56]. In addition to reduced oocyte quality, excessive aging can lead to slower passage through the developmental phases, as indicated in pigs [57]. SIRT1 is a protein whose activity is strongly associated with states of elevated concentrations of reactive oxygen species and has been proven to mitigate oocyte aging [58], and loss of SIRT1 causes increased oocyte aging [59]. SIRT1 engages in DNA repair processes by deacetylating key enzymes for substrate modification [60]. Engagement in this process can lead to depletion of NAD+ stores and lead to SIRT1 deficiency in oocytes [61,62]. It was indicated that mice and their DNA overexpressing SIRT1 were significantly more efficiently protected from cancer and repaired. It was also confirmed that SIRT1 is essential for proper telomere elongation and homeostasis [63,64]. In addition, deficiencies of the precursor (NAD+) for SIRT1 production are noted in older mouse ovaries [65]. In addition, there is a strong relationship between SIRT1, erythroid-related nuclear factor 2, and cyclin B1 on metaphase plate formation and thus meiotic division [66]. The NrF2 factor is thought to be an important protective factor against oxidative stress, and dysfunction has been linked to an upset in the rate of cell growth [67]. Deprivation of SIRT1 or nRf2 causes disturbances in their formation and also problems with entering the M phase in the oocyte aging [66].

The negative effects of aging can be counteracted by using SIRT1 stimulators. Some molecules have the natural ability to activate SIRT1 [68]. The positive effects of melatonin on oocytes, which activate several antioxidant pathways, may be one solution to excessive aging [69,70]. Among the activated genes associated with reducing ROS (Sod1, Sod2, Nrf2), SIRT1 is also prominent [8,71]. In contrast, restoration of mitochondrial function involves the elimination of accumulated reactive oxygen species to inhibit apoptosis–SIRT1/Sod2 [38]. Increasing attention is now being paid to the effect of heat stress, which can also be responsible for the accumulation of ROS in oocytes. However, this effect can be reversed by supplementation with nicotinamide mononucleotide (NMN) [72]. It is one of the precursor forms in the sirtuin formation pathway and can promote SIRT1 growth. Improvements were also obtained after nicotinamide mononucleotide supplementation in mice, yielding better quality oocytes, normal metaphase plate appearance, and greater fertilization success [73].

Aging has been correlated with progressive DNA hypomethylation. It has been indicated that the effects of dietary restrictions are directly related to changes in telomeres [74]. Recent evidence suggests that changes in methylation are mediated by SIRT1 [74,75]. However, no studies indicate the exact process by which SIRT1 increases methylation in CpG islands [76]. Hypermethylation may also serve to silence damaged strand fragments. It has been indicated that SIRT1, in cooperation with PcG and EZH2 proteins, can short-circuit chromatin condensation through hypoacetylation of H4K16, H3K9me2, and H3K27me3 (Figure 2). It has also been suggested that in acute injury states, the localization of methyltransferase 1 is associated with the presence of large amounts of SIRT1 [76].

## 5. Conclusions

In summary, studies in the past decade have shown the potential for resveratrol treatments to impact the quality of mammalian oocytes. Furthermore, resveratrol seems to be a compound which could be used to recover fertility with regard to aging-related sub- or infertility in some animal models. Nevertheless, the mechanism of action and how these treatments impact the human oocyte remains elusive, which underlines the great need to conduct new research in this regard. The effects of sirtuin 1 action or deficiency observed in the most important elements of the mammalian oocyte and during the most important processes illustrate its crucial importance in a multifaceted way, especially based on the fact that the proper alignment of the metaphase plate during oocyte maturation significantly impacts the fertilization success. This may also be of particular importance for pathologies of the female reproductive system, such as PCOS. To develop an adequate antiaging therapy for oocytes of advanced-age donors, it would be essential to establish a strategy which uses resveratrol at low and safe concentrations to activate the Sirtuin 1. All in all, this review aimed to stimulate such new research ideas and projects to underpin the cornerstone status of resveratrol and Sirtuin 1 for the aging process of the mammalian oocyte to extend both their life span and health span.

## Figures and Tables

**Figure 1 nutrients-14-05101-f001:**
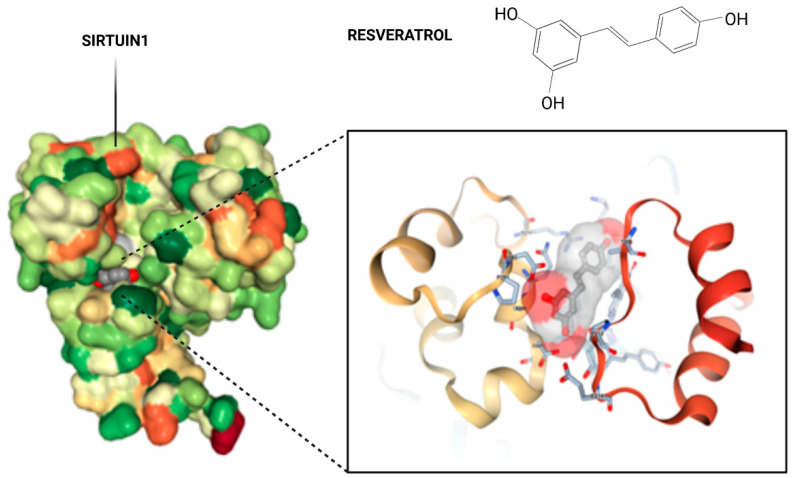
Crystal structure of Sirt1 in complex with resveratrol. The overall structure of SIRT1 with the protein domains and the catalytical core of the protein in magnification in amino acid structure binding to resveratrol. A pair of SIRT1 residues involved in interdomain interactions are shown in a stick model, with hydrogen bonds.

**Figure 2 nutrients-14-05101-f002:**
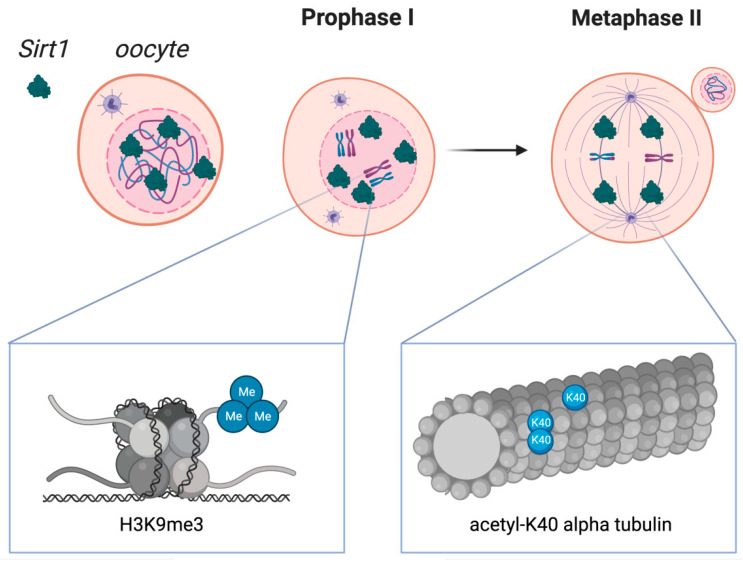
Sirtuin 1 distribution in oocytes during crucial stages of the first and second meiotic diviScheme 1. localization in oocytes at prophase I and at metaphase II. SIRT1 is mainly expressed in prophase I-arrested oocytes at nuclear localization and on the spindle in metaphase II oocytes. Molecular functions of Sirtuin 1 during meiotic divisions are shown at a chromatin level (left box) and at the spindle formation level (right box). Heterochromatin formation with possible impact on chromosome compaction, gene silencing, and DNA damage repair signaling have been reported upon SIRT1, which has shown to promote the H3K9me3 deposition, as depicted in the left zoomed-in box. As shown in the right zoomed-in box, SIRT1 also regulates the acetylation of α-tubulin, therefore influencing the formation of acetyl K-40 α-tubulin and suggesting an impact during spindle alignment.

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
