# Peer review of "Resveratrol and SIRT1: Antiaging Cornerstones for Oocytes?"

_nutrients, 2022, doi:10.3390/nu14235101_

Round 1

Reviewer 1 Report

This review aims to elucidate the role, influence and interplay of resveratrol and SIRT1 on mammalian oocyte quality in the reproductive aging context.

Despite the interesting topic, authors just summarized the original papers without deep understanding, expecially in regard to resveratrol; indeed, resveratrol action/role is largely lacking in the narrative as well as its crosstalk with Sirt1. Both text and figures are poor and fail to summarize key aspects related to the topic.

All figures are not linked to the main text and they only provide general informations. Authors should then include and describe in the text all the mechanisms shown in the figures.  Specifically,

Figure 1, "Crystal structure of Sirt1 in complex with resveratrol", which part describes this complex/interaction between Sirt1 and Res in the text? 

Figure 2, "Effects of resveratrol on ovaries and endometrium", all mechanisms depicted in this figure are not described in the main text.

Figure 3, also in this case, all mechanisms depicted in the figure are not described in the main text.

Based on which data authors make their conclusions?

Indeed, author stated, "It can be concluded, that recent studies have shown the potential of resveratrol treatments to restore the quality of aged oocytes in different mammalian species", but the resveratrol's action is poorly mentioned and without an exhaustive description.

Author Response

Dear  Editors,

Dear Reviewer,

Thank you for inviting us to respond to the very thoughtful and constructive reviewer comments. We greatly appreciate the reviewers time and believe our revised manuscript has become more well-rounded as a result.

We have incorporated all suggestions throughout the manuscript and they are highlighted in red. Below is a point-by-point response to reviewers’ comments to clarify which edits were made.

We are happy to respond to additional requests if they arise.

Sincerely,

Paweł Kordowitzki

Please note our following explanations:

Detailed answers to reviewer 1

REVIEWER: "Despite the interesting topic, authors just summarized the original papers without deep understanding, expecially in regard to resveratrol; indeed, resveratrol action/role is largely lacking in the narrative as well as its crosstalk with Sirt1. Both text and figures are poor and fail to summarize key aspects related to the topic."

ANSWER: We thank the reviewer for his/her erudite comment, which we appreciate. In response, we now acknowledge this point, and we have added more information to the entire manuscript. A new stion on the interaction between SIRT1 and resveratrol as been added,as follows:

  1. Interaction between SIRT1 and Resveratrol

The enzyme consists of N- and C-domain extensions and a catalytic core [15]. The catalytic domain is formed by the Rossmann fold (components: β1, β2, β6-9 and α1, α7, α8, α12) and the subdomain responsible for zinc binding (α10, α11), the former of which is the dominant subdomain in the catalytic core of sirtuin 1. The latter, on the other hand, which is smaller, is the result of the fusion of two insertions of the origin of the NAD+ linking domain [16]. The substrate that attaches to the active site, between these subdomains, forces a change in the position of the Rossmann fold and the second domain, resulting in the ability to attach to the sirtuin NAD+. This is defined as a transition of the SIRT1 form from open to closed, and the hydrophobic interdomain groove is then observed to disappear and the minor subdomain binds to zinc [Zn2+] [17,18]. Various molecules that modulate the activity of the enzyme can bind to the active site. Inauhzin, a SIRT1 inhibitor that was used to limit tumor growth, bound to the active center via hydrophobic bonds and was positioned deep in the interdomain groove between the active site and peripherally attached NAD+ [19]. In the case of EX527, a potent indole inhibitor of SIRT1, it was indicated that interaction with the active site also contributes to changing the conformation of NAD+ and sealing the catalytic pocket, and inducing blockade of acetylated lysine access to the substrate in deacetylation reactions [20]. In contrast, resveratrol allosterically modifies the activity of the SIRT1. Resveratrol has shown to bind to the catalytical core of Sirtuin1 (Fig.1). In the form of three molecules (Res.1, Res.2, and Res.3), it binds in two domains: Res.1 and Res.2 bind to the N-terminal domain (NTD) and Res.3 binds to the C-terminal domain [21]. It is worth noting that at the NTD site, resveratrol molecules also bind to peptides with which SIRT1 cooperates, where they facilitate, for example, bridging between proteins if only in the case of interaction with P53-AMC [22]. Resveratrol activity also facilitates phosphorylation at the C-domain by promoting the binding of SIRT1 to LKB1 (liver kinase B1), one of the components of biogenesis and mitochondrial respiration [23]. Therefore, resveratrol levels may be translated, via SIRT1, into the development of inflammation, regulation of apoptosis, mitochondrial function, and protection from oxidative stress. Among other reasons, sirtuin has become a therapeutic target in reproductive system diseases [24]. Mitochondrial activity, as well as the efficiency of the organization of cell organelles, is one of the keys to achieving good quality oocytes. Hence, the use of low-dose resveratrol in disease entities involving the ovaries and disorders related to oocyte development appears to be a reasonable approach.  

REVIEWER: "All figures are not linked to the main text and they only provide general informations. Authors should then include and describe in the text all the mechanisms shown in the figures. Specifically,Figure 1, "Crystal structure of Sirt1 in complex with resveratrol", which part describes this complex/interaction between Sirt1 and Res in the text?Figure 2, "Effects of resveratrol on ovaries and endometrium", all mechanisms depicted in this figure are not described in the main text.Figure 3, also in this case, all mechanisms depicted in the figure are not described in the main text."

ANSWER: We want to thank the reviewer for this comment, and  in response, we have corrected these issues. Figure 1 appears now after the newly added section 2 which describes the interactions. Furthermore we decided deleting figure2. And former figure 3 has be renamed to figure2. The processes described in figure 2 are mentioned in section 3 and 4 of the manuscript.

REVIEWER: "Based on which data authors make their conclusions?"

ANSWER: We want to thank the reviewer for this suggestion. We have rewritten the concluions based on the main text and new sections as follows:

In summary, studies in the past decade have shown the potential of resveratrol treatments to impact the quality of mammalian oocytes. Furhtermore, resveratrol seems to be compound which could be used to recover fertility with reagard to aging-related sub- or infertility in some animal models. Nevertheless, the mechanism of action and how these treatments impact the human oocyte remains elusive, what underlines the great need to conduct new research in this regard. The effects of sirtuin 1 action or deficiency observed in the most important elements of the mammalian oocyte and during the most important processes illustrate its crucial importance in a multifacetted way, especially based on the fact that the proper alignment of the metaphase plate during oocyte maturation significantly impacts the fertilization success. This may be of particular importance also for pathologies of the female reproductive system, such as PCOS. To devevlop an adequte anti-aging therapy for oocytes of advanced age donors, it would be essential to estabish a strategy which uses resveratrol at low and save concentrations to activate the Sirtuin 1 . All in all, this review aimed to stimulate such new reseach ideas and projects, to underpin the conerstone status of resveratrol and sirtuin 1 for the aging process of the mammalian oocyte, to extend both their life span and health span.

REVIEWER: "Indeed, author stated, "It can be concluded, that recent studies have shown the potential of resveratrol treatments to restore the quality of aged oocytes in different mammalian species", but the resveratrol's action is poorly mentioned and without an exhaustive description."

ANSWER: We want to thank the reviewer for his/her thorough review of our manuscript.  This is a very important point and in response, we have added the following sentences with regard to resveratrol additionally to the new section 2 which describes the action on Sirt1:

Resveratrol is a phytoalexin and plant polyphenol detected among others in red grapes, red wine, and some nuts [1]. Interestingly, it has been reported that resveratrol supplemented at specific low concentrations shows anti-aging properties and is able to enhance the activity of so called Sirtuins [1,2]. Furthermore, resveratrol is able to enhance mitochondrial metabolism, among others due to its impact on oxidative phosphorylation (OXPHOS) and mitochondrial biogenesis. However, it is important that to mention that the postive effects of resveratrol are present when this compound is used in low doses, since high-doses resveratrol could provoke the inhibition of P450 cytochromes, and could induce a mitochondrial-dependent cell death [2]. The supplementation with resveratrol as an antioxidant treatment to elevate oocyte quality and in vitro fertization outcome has been widely investigated in the past decade. Nevertheless, there is no clear consensus with regard to the mechanism of resveratrol’s action on human oocytes, despite their well-known influence on oocyte mitochondria [2]. Are more detailed description on the intereaction between resveratrol and SIRT1 will be discussed in section 2.

Reviewer 2 Report

In our opinion, the Grzeczka's proposal is too speculative. The proposal is not based on any conclusive result and are only approaches or opinions based on results  that although it is related, they do not allow a conclusion like the one proposed here. In fact,  the first sentence of conclusions "It can be concluded, that recent studies have shown the potential of resveratrol  treatments to restore the quality of aged oocytes in different mammalian species" is not based at all on what was discussed in the work. The work focuses more on repeat what is already published about SIRT 1 and resveratrol.

Author Response

Dear  Editors,

Dear Reviewer,

Thank you for inviting us to respond to the very thoughtful and constructive reviewer comments. We greatly appreciate the reviewers time and believe our revised manuscript has become more well-rounded as a result.

We have incorporated all suggestions throughout the manuscript and they are highlighted in red. Below is a point-by-point response to reviewers’ comments to clarify which edits were made.

We are happy to respond to additional requests if they arise.

Sincerely,

Paweł Kordowitzki

Please note our following explanations:

Detailed answers to reviewer 2

REVIEWER: "In our opinion, the Grzeczka's proposal is too speculative. The proposal is not based on any conclusive result and are only approaches or opinions based on results  that although it is related, they do not allow a conclusion like the one proposed here. In fact,  the first sentence of conclusions "It can be concluded, that recent studies have shown the potential of resveratrol  treatments to restore the quality of aged oocytes in different mammalian species" is not based at all on what was discussed in the work. The work focuses more on repeat what is already published about SIRT 1 and resveratrol."

ANSWER: We thank the reviewer for his/her erudite comment, which we appreciate. In response, we now acknowledge this point, and we have added more information to the entire manuscript. The conclusions have been rewrittwn, and a new section on the interaction between SIRT1 and resveratrol as been added,as follows:

Resveratrol is a phytoalexin and plant polyphenol detected among others in red grapes, red wine, and some nuts [1]. Interestingly, it has been reported that resveratrol supplemented at specific low concentrations shows anti-aging properties and is able to enhance the activity of so called Sirtuins [1,2]. Furthermore, resveratrol is able to enhance mitochondrial metabolism, among others due to its impact on oxidative phosphorylation (OXPHOS) and mitochondrial biogenesis. However, it is important that to mention that the postive effects of resveratrol are present when this compound is used in low doses, since high-doses resveratrol could provoke the inhibition of P450 cytochromes, and could induce a mitochondrial-dependent cell death [2]. The supplementation with resveratrol as an antioxidant treatment to elevate oocyte quality and in vitro fertization outcome has been widely investigated in the past decade. Nevertheless, there is no clear consensus with regard to the mechanism of resveratrol’s action on human oocytes, despite their well-known influence on oocyte mitochondria [2]. Are more detailed description on the intereaction between resveratrol and SIRT1 will be discussed in section 2.

2. Interaction between SIRT1 and Resveratrol

The enzyme consists of N- and C-domain extensions and a catalytic core [15]. The catalytic domain is formed by the Rossmann fold (components: β1, β2, β6-9 and α1, α7, α8, α12) and the subdomain responsible for zinc binding (α10, α11), the former of which is the dominant subdomain in the catalytic core of sirtuin 1. The latter, on the other hand, which is smaller, is the result of the fusion of two insertions of the origin of the NAD+ linking domain [16]. The substrate that attaches to the active site, between these subdomains, forces a change in the position of the Rossmann fold and the second domain, resulting in the ability to attach to the sirtuin NAD+. This is defined as a transition of the SIRT1 form from open to closed, and the hydrophobic interdomain groove is then observed to disappear and the minor subdomain binds to zinc [Zn2+] [17,18]. Various molecules that modulate the activity of the enzyme can bind to the active site. Inauhzin, a SIRT1 inhibitor that was used to limit tumor growth, bound to the active center via hydrophobic bonds and was positioned deep in the interdomain groove between the active site and peripherally attached NAD+ [19]. In the case of EX527, a potent indole inhibitor of SIRT1, it was indicated that interaction with the active site also contributes to changing the conformation of NAD+ and sealing the catalytic pocket, and inducing blockade of acetylated lysine access to the substrate in deacetylation reactions [20]. In contrast, resveratrol allosterically modifies the activity of the SIRT1. Resveratrol has shown to bind to the catalytical core of Sirtuin1 (Fig.1). In the form of three molecules (Res.1, Res.2, and Res.3), it binds in two domains: Res.1 and Res.2 bind to the N-terminal domain (NTD) and Res.3 binds to the C-terminal domain [21]. It is worth noting that at the NTD site, resveratrol molecules also bind to peptides with which SIRT1 cooperates, where they facilitate, for example, bridging between proteins if only in the case of interaction with P53-AMC [22]. Resveratrol activity also facilitates phosphorylation at the C-domain by promoting the binding of SIRT1 to LKB1 (liver kinase B1), one of the components of biogenesis and mitochondrial respiration [23]. Therefore, resveratrol levels may be translated, via SIRT1, into the development of inflammation, regulation of apoptosis, mitochondrial function, and protection from oxidative stress. Among other reasons, sirtuin has become a therapeutic target in reproductive system diseases [24]. Mitochondrial activity, as well as the efficiency of the organization of cell organelles, is one of the keys to achieving good quality oocytes. Hence, the use of low-dose resveratrol in disease entities involving the ovaries and disorders related to oocyte development appears to be a reasonable approach.  

5.Conlusions

In summary, studies in the past decade have shown the potential of resveratrol treatments to impact the quality of mammalian oocytes. Furhtermore, resveratrol seems to be compound which could be used to recover fertility with reagard to aging-related sub- or infertility in some animal models. Nevertheless, the mechanism of action and how these treatments impact the human oocyte remains elusive, what underlines the great need to conduct new research in this regard. The effects of sirtuin 1 action or deficiency observed in the most important elements of the mammalian oocyte and during the most important processes illustrate its crucial importance in a multifacetted way, especially based on the fact that the proper alignment of the metaphase plate during oocyte maturation significantly impacts the fertilization success. This may be of particular importance also for pathologies of the female reproductive system, such as PCOS. To devevlop an adequte anti-aging therapy for oocytes of advanced age donors, it would be essential to estabish a strategy which uses resveratrol at low and save concentrations to activate the Sirtuin 1 . All in all, this review aimed to stimulate such new reseach ideas and projects, to underpin the conerstone status of resveratrol and sirtuin 1 for the aging process of the mammalian oocyte, to extend both their life span and health span.

Reviewer 3 Report

This article does not introduce the related knowledge of resveratrol and the research progress of the relationship between resveratrol and oocytes。

Author Response

Dear Editors and Reviewers,

Thank you for inviting us to respond to the very thoughtful and constructive reviewer comments. We greatly appreciate the reviewers time and believe our revised manuscript has become more well-rounded as a result.

We have incorporated all suggestions throughout the manuscript and they are highlighted in red. Below is a point-by-point response to reviewers’ comments to clarify which edits were made.

We are happy to respond to additional requests if they arise.

Sincerely,

Paweł Kordowitzki

Please note our following explanations:

Detailed answers to reviewer 3

REVIEWER: "This article does not introduce the related knowledge of resveratrol and the research progress of the relationship between resveratrol and oocytes"

ANSWER: We thank the reviewer for his/her erudite comment, which we appreciate. In response, we now acknowledge this point, and we have added more information to the entire manuscript. The conclusions have been rewritten, and a new section on the interaction between SIRT1 and resveratrol as been added,as follows:

Resveratrol is a phytoalexin and plant polyphenol detected among others in red grapes, red wine, and some nuts [1]. Interestingly, it has been reported that resveratrol supplemented at specific low concentrations shows anti-aging properties and is able to enhance the activity of so called Sirtuins [1,2]. Furthermore, resveratrol is able to enhance mitochondrial metabolism, among others due to its impact on oxidative phosphorylation (OXPHOS) and mitochondrial biogenesis. However, it is important that to mention that the postive effects of resveratrol are present when this compound is used in low doses, since high-doses resveratrol could provoke the inhibition of P450 cytochromes, and could induce a mitochondrial-dependent cell death [2]. The supplementation with resveratrol as an antioxidant treatment to elevate oocyte quality and in vitro fertization outcome has been widely investigated in the past decade. Nevertheless, there is no clear consensus with regard to the mechanism of resveratrol’s action on human oocytes, despite their well-known influence on oocyte mitochondria [2]. Are more detailed description on the intereaction between resveratrol and SIRT1 will be discussed in section 2.

2. Interaction between SIRT1 and Resveratrol

The enzyme consists of N- and C-domain extensions and a catalytic core [15]. The catalytic domain is formed by the Rossmann fold (components: β1, β2, β6-9 and α1, α7, α8, α12) and the subdomain responsible for zinc binding (α10, α11), the former of which is the dominant subdomain in the catalytic core of sirtuin 1. The latter, on the other hand, which is smaller, is the result of the fusion of two insertions of the origin of the NAD+ linking domain [16]. The substrate that attaches to the active site, between these subdomains, forces a change in the position of the Rossmann fold and the second domain, resulting in the ability to attach to the sirtuin NAD+. This is defined as a transition of the SIRT1 form from open to closed, and the hydrophobic interdomain groove is then observed to disappear and the minor subdomain binds to zinc [Zn2+] [17,18]. Various molecules that modulate the activity of the enzyme can bind to the active site. Inauhzin, a SIRT1 inhibitor that was used to limit tumor growth, bound to the active center via hydrophobic bonds and was positioned deep in the interdomain groove between the active site and peripherally attached NAD+ [19]. In the case of EX527, a potent indole inhibitor of SIRT1, it was indicated that interaction with the active site also contributes to changing the conformation of NAD+ and sealing the catalytic pocket, and inducing blockade of acetylated lysine access to the substrate in deacetylation reactions [20]. In contrast, resveratrol allosterically modifies the activity of the SIRT1. Resveratrol has shown to bind to the catalytical core of Sirtuin1 (Fig.1). In the form of three molecules (Res.1, Res.2, and Res.3), it binds in two domains: Res.1 and Res.2 bind to the N-terminal domain (NTD) and Res.3 binds to the C-terminal domain [21]. It is worth noting that at the NTD site, resveratrol molecules also bind to peptides with which SIRT1 cooperates, where they facilitate, for example, bridging between proteins if only in the case of interaction with P53-AMC [22]. Resveratrol activity also facilitates phosphorylation at the C-domain by promoting the binding of SIRT1 to LKB1 (liver kinase B1), one of the components of biogenesis and mitochondrial respiration [23]. Therefore, resveratrol levels may be translated, via SIRT1, into the development of inflammation, regulation of apoptosis, mitochondrial function, and protection from oxidative stress. Among other reasons, sirtuin has become a therapeutic target in reproductive system diseases [24]. Mitochondrial activity, as well as the efficiency of the organization of cell organelles, is one of the keys to achieving good quality oocytes. Hence, the use of low-dose resveratrol in disease entities involving the ovaries and disorders related to oocyte development appears to be a reasonable approach.

5.Conlusions

In summary, studies in the past decade have shown the potential of resveratrol treatments to impact the quality of mammalian oocytes. Furhtermore, resveratrol seems to be compound which could be used to recover fertility with reagard to aging-related sub- or infertility in some animal models. Nevertheless, the mechanism of action and how these treatments impact the human oocyte remains elusive, what underlines the great need to conduct new research in this regard. The effects of sirtuin 1 action or deficiency observed in the most important elements of the mammalian oocyte and during the most important processes illustrate its crucial importance in a multifacetted way, especially based on the fact that the proper alignment of the metaphase plate during oocyte maturation significantly impacts the fertilization success. This may be of particular importance also for pathologies of the female reproductive system, such as PCOS. To devevlop an adequte anti-aging therapy for oocytes of advanced age donors, it would be essential to estabish a strategy which uses resveratrol at low and save concentrations to activate the Sirtuin 1 . All in all, this review aimed to stimulate such new reseach ideas and projects, to underpin the conerstone status of resveratrol and sirtuin 1 for the aging process of the mammalian oocyte, to extend both their life span and health span.

Reviewer 4 Report

In their manuscript, the authors summarized the latest findings related to resveratrol, SIRT1, their cross-talk and influence on the mammalian oocyte. While the topic is indeed interesting, the following questions need to be addressed:

-Figure 1 (Crystal structure of Sirt1 in complex with resveratrol) seems to be not inserted at the appropriate place in the text, because Sirt1 is not mentioned in the first paragraph of ‘Introduction’. Besides, the interaction between Sirt1 and resveratrol shown in figure1 was not described in this manuscript.

-I will suggest that the authors add the chemical structure of resveratrol in Figure 1 instead of Figure 2.

-The crystal structure of Sirt1 binds to resveratrol requires the reference cited.

-It is suggested to add labels (such as A and B) to the two figures in Figure 1 to distinguish.

- Figure 2 is not cited in the text.

- The author has well summarized the relationship between SIRT1 and quality of oocyte as well as SIRT1 and aging. However, as an important topic to be discussed in this text, the summary of the relationship between resveratrol and SIRT1 is relatively lacking. I will suggest the authors add more relevant summaries and discussions to the text.

Author Response

Dear Editors and Reviewers,

Thank you for inviting us to respond to the very thoughtful and constructive reviewer comments. We greatly appreciate the reviewers time and believe our revised manuscript has become more well-rounded as a result.

We have incorporated all suggestions throughout the manuscript and they are highlighted in red. Below is a point-by-point response to reviewers’ comments to clarify which edits were made.

We are happy to respond to additional requests if they arise.

Sincerely,

Paweł Kordowitzki

Please note our following explanations:

Detailed answers to reviewer 4

REVIEWER: "Figure 1 (Crystal structure of Sirt1 in complex with resveratrol) seems to be not inserted at the appropriate place in the text, because Sirt1 is not mentioned in the first paragraph of ‘Introduction’. Besides, the interaction between Sirt1 and resveratrol shown in figure1 was not described in this manuscript.I will suggest that the authors add the chemical structure of resveratrol in Figure 1 instead of Figure 2.The crystal structure of Sirt1 binds to resveratrol requires the reference cited. It is suggested to add labels (such as A and B) to the two figures in Figure 1 to distinguish. Figure 2 is not cited in the text."

ANSWER: We want to thank the reviewer for this comment, and in response, we have corrected these issues. Figure 1 appears now after the newly added section 2 which describes the interactions between Resveratrol and Sirt1, and the citation has been added to the figure legend. Figure 1 has be re-drawn and appears now in a clar order. Furthermore, we decided deleting figure 2.

REVIEWER:"The author has well summarized the relationship between SIRT1 and quality of oocyte as well as SIRT1 and aging. However, as an important topic to be discussed in this text, the summary of the relationship between resveratrol and SIRT1 is relatively lacking. I will suggest the authors add more relevant summaries and discussions to the text."

ANSWER: We thank the reviewer for his/her erudite comment, which we appreciate. In response, we now acknowledge this point, and we have added more information to the entire manuscript. The conclusions have been rewritten, and a new section on the interaction between SIRT1 and resveratrol as been added,as follows:

Resveratrol is a phytoalexin and plant polyphenol detected among others in red grapes, red wine, and some nuts [1]. Interestingly, it has been reported that resveratrol supplemented at specific low concentrations shows anti-aging properties and is able to enhance the activity of so called Sirtuins [1,2]. Furthermore, resveratrol is able to enhance mitochondrial metabolism, among others due to its impact on oxidative phosphorylation (OXPHOS) and mitochondrial biogenesis. However, it is important that to mention that the postive effects of resveratrol are present when this compound is used in low doses, since high-doses resveratrol could provoke the inhibition of P450 cytochromes, and could induce a mitochondrial-dependent cell death [2]. The supplementation with resveratrol as an antioxidant treatment to elevate oocyte quality and in vitro fertization outcome has been widely investigated in the past decade. Nevertheless, there is no clear consensus with regard to the mechanism of resveratrol’s action on human oocytes, despite their well-known influence on oocyte mitochondria [2]. Are more detailed description on the intereaction between resveratrol and SIRT1 will be discussed in section 2.

2. Interaction between SIRT1 and Resveratrol

The enzyme consists of N- and C-domain extensions and a catalytic core [15]. The catalytic domain is formed by the Rossmann fold (components: β1, β2, β6-9 and α1, α7, α8, α12) and the subdomain responsible for zinc binding (α10, α11), the former of which is the dominant subdomain in the catalytic core of sirtuin 1. The latter, on the other hand, which is smaller, is the result of the fusion of two insertions of the origin of the NAD+ linking domain [16]. The substrate that attaches to the active site, between these subdomains, forces a change in the position of the Rossmann fold and the second domain, resulting in the ability to attach to the sirtuin NAD+. This is defined as a transition of the SIRT1 form from open to closed, and the hydrophobic interdomain groove is then observed to disappear and the minor subdomain binds to zinc [Zn2+] [17,18]. Various molecules that modulate the activity of the enzyme can bind to the active site. Inauhzin, a SIRT1 inhibitor that was used to limit tumor growth, bound to the active center via hydrophobic bonds and was positioned deep in the interdomain groove between the active site and peripherally attached NAD+ [19]. In the case of EX527, a potent indole inhibitor of SIRT1, it was indicated that interaction with the active site also contributes to changing the conformation of NAD+ and sealing the catalytic pocket, and inducing blockade of acetylated lysine access to the substrate in deacetylation reactions [20]. In contrast, resveratrol allosterically modifies the activity of the SIRT1. Resveratrol has shown to bind to the catalytical core of Sirtuin1 (Fig.1). In the form of three molecules (Res.1, Res.2, and Res.3), it binds in two domains: Res.1 and Res.2 bind to the N-terminal domain (NTD) and Res.3 binds to the C-terminal domain [21]. It is worth noting that at the NTD site, resveratrol molecules also bind to peptides with which SIRT1 cooperates, where they facilitate, for example, bridging between proteins if only in the case of interaction with P53-AMC [22]. Resveratrol activity also facilitates phosphorylation at the C-domain by promoting the binding of SIRT1 to LKB1 (liver kinase B1), one of the components of biogenesis and mitochondrial respiration [23]. Therefore, resveratrol levels may be translated, via SIRT1, into the development of inflammation, regulation of apoptosis, mitochondrial function, and protection from oxidative stress. Among other reasons, sirtuin has become a therapeutic target in reproductive system diseases [24]. Mitochondrial activity, as well as the efficiency of the organization of cell organelles, is one of the keys to achieving good quality oocytes. Hence, the use of low-dose resveratrol in disease entities involving the ovaries and disorders related to oocyte development appears to be a reasonable approach.

5.Conlusions

In summary, studies in the past decade have shown the potential of resveratrol treatments to impact the quality of mammalian oocytes. Furhtermore, resveratrol seems to be compound which could be used to recover fertility with reagard to aging-related sub- or infertility in some animal models. Nevertheless, the mechanism of action and how these treatments impact the human oocyte remains elusive, what underlines the great need to conduct new research in this regard. The effects of sirtuin 1 action or deficiency observed in the most important elements of the mammalian oocyte and during the most important processes illustrate its crucial importance in a multifacetted way, especially based on the fact that the proper alignment of the metaphase plate during oocyte maturation significantly impacts the fertilization success. This may be of particular importance also for pathologies of the female reproductive system, such as PCOS. To devevlop an adequte anti-aging therapy for oocytes of advanced age donors, it would be essential to estabish a strategy which uses resveratrol at low and save concentrations to activate the Sirtuin 1 . All in all, this review aimed to stimulate such new reseach ideas and projects, to underpin the conerstone status of resveratrol and sirtuin 1 for the aging process of the mammalian oocyte, to extend both their life span and health span.

Round 2

Reviewer 1 Report

The manuscript has been notably improved with the last editing. The following minor revisions are required.

• All statements from line 27 to line 39 are not supported by appropriate references; therefore, some references need to be included.

• Add the reference PMID: 21191486 (line 48) in regard to the dose-dependency positive effects of resveratrol, and the reference PMID: 25656643 (line 49) in regard to the mitochondrial-dependent cell death.

• Line 54-55, the following sentence needs a revision "Are more detailed description on the intereaction between resveratrol and SIRT1 will be discussed in section 2".

• There are several typing errors, e.g.  "furhtermore" (line 264), "reagard" (line 265), "multifacetted" (line 271), "To devevlop an adequte" (line 275), " estabish" (line 276) etc. Please revised all the text.

Author Response

All comments have been addressed.

Reviewer 2 Report

The manuscript was consistently improved. 

Author Response

All comments have been addressed.

Reviewer 3 Report

accept

Author Response

All comments have been addressed.
